# OpenReview forum: "Revoking Amnesia: RL-based Trajectory Optimization to Resurrect Erased Concepts in Diffusion Models"
_ICLR.cc/2026/Conference — ICLR 2026 Conference Withdrawn Submission_

### Official Review · Reviewer_PtnU · 2025-10-17

**Soundness:** 3
**Presentation:** 3
**Contribution:** 2
**Rating:** 2
**Confidence:** 4

**Summary:**

The paper presents an approach to recover erased concepts from diffusion models. They propose to do this using a GRPO based trajectory optimization algorithm. The authors compare their approach with UnlearnDiffAtk, Ring-A-Bell and R2A on attacking the following erasure methods - AC, ESD, EA and EAP showing promising results.

**Strengths:**

- The approach is sound and intuitive and shows promising results in various settings.
- The paper is generally well written and easy to follow and understand. I especially like the figures and visuals -- they are easy to understand and clearly convey the message.

**Weaknesses:**

- The threat model here is unclear to me. The authors try to make a distinction with prompt based adversarial techniques by saying this - "Unlike prompt-based approaches that search vast spaces of discrete tokens or high-dimensional embeddings, we pose recovery in a compact, interpretable action space". However, this explanation is insufficient. The proposed approach requires complete white box access to the weights AND control over the entire denoising trajectory. Many prompt based techniques such as Ring-A-Bell are black box attacks.
- The authors propose using "K" rewards models to optimize the trajectory including erased class specific reward model. This is limiting since an adversary will first need to collect images, train a classification model and then do the trajectory optimization.
- I find the following statement authors very surprising "erasure techniques developed for SD exhibit severely degraded performance when applied to these modern systems." Can you please add a citation for this? What evidence is this claim based on?
- The experimental evaluations are lacking. It needs to be compared with more baseline erasure methods as well as red teaming methods. Such as how does it compare to MMA-Diffusion [1] and P4D. How does it perform against more recent defenses which specifically propose defences against adversarial attacks such as AdvLearn [2]?
- The method is only evaluated against weight update based erasure techniques. How well does this apply to inference based erasure techniques such as TraSCE, Safe-Latent-Diffusion, if at all?
- Many relevant citations are missing.

[1] Yang, Yijun, et al. "Mma-diffusion: Multimodal attack on diffusion models." Proceedings of the IEEE/CVF Conference on Computer Vision and Pattern Recognition. 2024

[2] Yimeng Zhang, Xin Chen, Jinghan Jia, Yihua Zhang, Chongyu Fan, Jiancheng Liu, Mingyi Hong, Ke Ding, and Sijia Liu. Defensive unlearning with adversarial training for robust concept erasure in diffusion models.

**Questions:**

- Can the authors clarify and clearly state the threat model they are considering in this paper and position it against previous prompt based attacks. If and what additional assumption are they making for this attack to be successful in a practical setting?
- Can the authors also clarify based on their threat model would a simple adversarial search in the VAE latent space using a reward model not be effective? This wouldn't even assume access to the U-Net model.

---

### Official Review · Reviewer_DUKH · 2025-10-29

**Soundness:** 3
**Presentation:** 4
**Contribution:** 3
**Rating:** 6
**Confidence:** 3

**Summary:**

This paper investigates the limitations of current machine unlearning methods (e.g., ESD, UCE), demonstrating through experimental analysis that these approaches do not truly erase the target concept. Instead, they achieve unlearning by perturbing the diffusion model’s sampling trajectory associated with that concept. Therefore, the authors propose RevAM, an attack framework that optimizes the diffusion sampling trajectory to recover previously erased concepts. Experimental results show that RevAM achieves significantly higher attack success rates while reducing computational time by up to 10× compared to baselines.

**Strengths:**

1. The paper provides valuable insights by experimentally showing that existing unlearning methods primarily modify diffusion trajectories rather than eliminating concept representations, which opens up a new perspective for designing more effective unlearning approaches.
2. The proposed RevAM achieves superior attack success rates and efficiency, demonstrating both stronger performance and lower computational cost compared to competing methods.

**Weaknesses:**

1. As shown in Table 1 and Table 6, the AC unlearning method minimally alters the diffusion trajectory, yet in Table 3, RevAM’s attack success rate against AC is higher than that against SED or EA — this contradiction requires clarification.
2. The evaluation is incomplete: there is no analysis of how the optimized model affects unrelated concepts (e.g., optimizing “nudity” unlearning might unintentionally impair the ability to generate “man”).
3. In the main experiments, beyond attack success rates, additional evaluation metrics such as CLIP Score should be reported to measure semantic alignment and generation quality.

**Questions:**

1. The paper lacks an ablation study or justification for the chosen ranges of $\rho$ and $\phi$.
2. The selection of baseline unlearning methods omits text-level eraser methods, such as Latent Guard [1] and SoftPrompt [2], which focus on concept removal via textual conditioning.
3. It is unclear what the CLIP Score in Table 5 represents. If it measures text–image alignment, the reported values appear unexpectedly high.
4. Since Table 5 evaluates on celebrity-related concepts, using a Face Similarity metric would be more appropriate than CLIP Score.

Minor Issues:
The caption of Table 1 contains some formatting or clarity issues that should be corrected.

[1] Liu R, Khakzar A, Gu J, et al. Latent guard: a safety framework for text-to-image generation[C]//European Conference on Computer Vision.

[2] Yuan L, Li X, Xu C, et al. Promptguard: Soft prompt-guided unsafe content moderation for text-to-image models[J]. arXiv preprint arXiv:2501.03544, 2025.

---

### Official Review · Reviewer_5h5S · 2025-11-02

**Soundness:** 2
**Presentation:** 2
**Contribution:** 2
**Rating:** 2
**Confidence:** 4

**Summary:**

The paper studies how to recover unlearned concepts at inference time. The high-level idea is to redirect the guidance signal in the denoising process toward the unlearned region. The authors formulate this as a reinforcement learning problem and adapt Group Relative Policy Optimization (GRPO) to better guide the generation process.

**Strengths:**

•  The problem of recovering unlearned concepts studied in this paper is important.

•  The experiments were conducted on Flux, a modern and powerful architecture.

•  The results look promising.

**Weaknesses:**

Limitations in Writing and Motivation

- The claim that changing the architecture (from U-Net to Flux) degrades unlearning performance seems weak and lacks supporting evidence. For example, ESD and its variants are, in principle, architecture-agnostic.

- The argument that “the limitations of concept unlearning on model architecture led to a new angle: sampling dynamics” is also weak and not clearly related. A better example could be drawn from [1] or more recently [2], where the authors showed that confusing gradients can lead to similar effects.

- The main motivation of this paper is weakly supported and resembles previous work [2].

- In my opinion, the Introduction should be rewritten to make the motivation of the proposed method clearer, rather than focusing on difficulties or architectural changes.

- Measuring cosine similarity and the norm difference between two vectors/predictions is insufficient to support the claim that “concept unlearning actually steers the denoising trajectory away from the target region.”

- Regarding Flow Matching and Diffusion Models, while they share a similar principle—transforming data distribution to a prior distribution through forward and reverse processes—the paper should describe this relationship more clearly to avoid confusion. The authors seem to conflate traditional diffusion models such as DDPM and DDIM (which rely on log gradients or score functions) with flow-matching-based models (which rely on velocity fields).

- Section 4.3 should be rewritten as an introductory paragraph for GRPO for Velocity Policy Optimization (Section 4.4) rather than as a separate section.

- Section 4.4 is also poorly written. Key terminologies such as M_sub, π_ref, etc., are not defined, making it difficult to follow.

Limitations of the method

- Why is GRPO used instead of DPO, as in other recent machine unlearning approaches [3]? The authors do not justify their choice of GRPO over simpler alternatives.

- The method requires K reward models (each designed for a specific unlearning task) and generates G roll-out images from the old policy model for each prompt. While the paper demonstrates faster successful attacks, it does not quantify the significant computational cost involved

[1] Pham, Minh, et al. "Circumventing concept erasure methods for text-to-image generative models." ICLR 2024

[2] Lu, Kevin, et al. "When Are Concepts Erased From Diffusion Models?." NeurIPS 2025.

[3] Zhang, Ruiqi, et al. "Negative preference optimization: From catastrophic collapse to effective unlearning." arXiv preprint arXiv:2404.05868 (2024).

**Questions:**

In addition to the weaknesses mentioned above, please address the following concerns:

-	How does this work compare to [2]? More specifically, in [2], the authors proposed two definitions of concept unlearning: destruction-based and guidance-based. Section 3 of this paper seems to correspond to the guidance-based definition in [2].

[2] Lu, Kevin, et al. "When Are Concepts Erased From Diffusion Models?." NeurIPS 2025.

---

### Note · Authors · 2025-11-12

**Comment:**

Thanks to all reviewers and Area Chairs for your diligent efforts. Regarding the initial scores, we have decided not to forgo the rebuttal phase.

Best regards.

**Withdrawal Confirmation:**

I have read and agree with the venue's withdrawal policy on behalf of myself and my co-authors.